# Newborn Screening for Pompe Disease

**DOI:** 10.3390/ijns6020031

**Published:** 2020-04-05

**Authors:** Takaaki Sawada, Jun Kido, Kimitoshi Nakamura

**Affiliations:** Department of Pediatrics, Graduate School of Medical Sciences, Kumamoto University, Kumamoto 860-8556, Japan; sawada.takaki@kuh.kumamoto-u.ac.jp (T.S.); nakamura@kumamoto-u.ac.jp (K.N.)

**Keywords:** Pompe disease, newborn screening, pseudodeficiency, genotype-phenotype correlation, treatment and follow-up

## Abstract

Glycogen storage disease type II (also known as Pompe disease (PD)) is an autosomal recessive disorder caused by defects in α-glucosidase (AαGlu), resulting in lysosomal glycogen accumulation in skeletal and heart muscles. Accumulation and tissue damage rates depend on residual enzyme activity. Enzyme replacement therapy (ERT) should be started before symptoms are apparent in order to achieve optimal outcomes. Early initiation of ERT in infantile-onset PD improves survival, reduces the need for ventilation, results in earlier independent walking, and enhances patient quality of life. Newborn screening (NBS) is the optimal approach for early diagnosis and treatment of PD. In NBS for PD, measurement of AαGlu enzyme activity in dried blood spots (DBSs) is conducted using fluorometry, tandem mass spectrometry, or digital microfluidic fluorometry. The presence of pseudodeficiency alleles, which are frequent in Asian populations, interferes with NBS for PD, and current NBS systems cannot discriminate between pseudodeficiency and cases with PD or potential PD. The combination of *GAA* gene analysis with NBS is essential for definitive diagnoses of PD. In this review, we introduce our experiences and discuss NBS programs for PD implemented in various countries.

## 1. Introduction

Glycogen storage disease type II (OMIM 232300), also known as Pompe disease (PD), is an autosomal recessive disorder caused by a defect in α-glucosidase (AαGlu; EC 3.2.1.20/3), resulting in the accumulation of lysosomal glycogen in skeletal and heart muscles [1]. The rates of accumulation and tissue damage depend on the residual enzyme activity. Patients with infantile-onset PD (IOPD) exhibit nearly complete absence of AαGlu activity and develop hypotonia and hypertrophic cardiomyopathy during infancy. Patients with IOPD eventually die of cardiorespiratory failure because massive amounts of glycogen accumulate in their skeletal and heart muscles. Patients with late-onset PD (LOPD) who exhibit marked reductions in AαGlu activity exhibit skeletal muscle dysfunction but rarely present with cardiac muscle disorders. The onset time and phenotypes of LOPD are variable, and patients are likely to exhibit manifestations in the fifth decade or later in life [2]. Enzyme replacement therapy (ERT) is essential for the treatment of IOPD [3,4]. ERT should be started before symptoms are clearly present, prior to the development of irreversible damage, to achieve optimal outcomes [5]. Early initiation of ERT can improve survival rates and quality of life in patients with IOPD, reducing the need for ventilation and leading to earlier independent walking [6].

Newborn screening (NBS) is an optimal approach for early diagnosis and treatment of IOPD. NBS, including pilot studies, is currently being carried out in several countries worldwide. Here, we describe NBS programs for PD, the diagnostic algorithm for PD, AαGlu enzyme assays using dried blood spots (DBSs) and fibroblasts, *GAA* gene analysis, and pseudodeficiency in *GAA*. Additionally, we discuss optimal treatments for PD, the current status of NBS worldwide, and future challenges in the development of NBS programs.

## 2. NBS Program for PD

NBS for PD is currently performed in Taiwan, Japan, and several states in the United States of America (USA). Current NBS systems measure AαGlu enzyme activity in DBSs.

In Japan, DBSs are prepared at maternity clinics or obstetrics departments using standard procedures at 4–6 days after birth for newborn mass screening according to public health system guidelines. After dropping blood spots onto filter papers (Toyo Roshi Kaisha, Ltd., Tokyo, Japan), DBSs are dried for at least 4 h at room temperature and sent to the Newborn Screening Center in Kumamoto within 1 week after preparation. The AαGlu activity in DBSs is then analyzed. The cutoff values in the AαGlu activity assay using DBSs differ for each research group but are set between 0.1 and 0.5 percentile values for the population or 20% to 30% of the mean value of the population. For newborns with values less than the cutoff values, a second AαGlu activity assay and *GAA* gene analysis are then performed [7].

At our institution, NBS for PD is performed in three steps (Figure 1). In the first step, newborns with AαGlu activity under the cutoff value of 6.5 pmol/h/disk (10% of the median value in the population) are recalled, and their DBSs are evaluated again. In the second step, using the Ba/Zn method, newborns with AαGlu activity under the cutoff value of 5.5 pmol/h/disk are called to the hospital within 2 months for a clinical examination. The infants are examined using physical and biochemical assays to confirm symptomatic signs of IOPD, and a third AαGlu assay is also performed. Finally, *GAA* gene analysis is performed in newborns with AαGlu activity under the cutoff value of 4.0 pmol/h/disk. The period after birth until the result of the first AαGlu assay is acquired is 1–2 weeks, and the period until the result of the second AαGlu assay is acquired is within 4 weeks. Thereafter, the period from birth until clinical examination is within 2 months, and the period from birth until *GAA* gene analysis and final diagnosis is up to 6 months [7].

A definitive diagnosis of PD is achieved in patients harboring two known pathogenic *GAA* variants with decreased AαGlu activity in the blood (leukocytes, DBSs, isolated lymphocytes) or another tissue, such as fibroblast. A probable diagnosis for PD can be made in patients with decreased enzyme activity but ambiguous *GAA* gene analysis owing to the presence of molecular variants of unknown significance (VOUS). Moreover, the prevalence of pseudodeficiency alleles is high in Asian populations.

Figure 2 shows the diagnostic algorithm for PD. Clinicians can definitively diagnose patients with IOPD if they present with certain clinical manifestations, including heart or skeletal muscle deficiencies. Patients with LOPD definitively diagnosed by *GAA* gene analysis will need to be regularly followed up for the development of signs or symptoms related to PD, even if their *GAA* gene variants are known, because there is considerable variation in how and when patients will present symptoms. Patients with one or no known variants exhibiting decreased enzymatic activity should receive additional tests, including physical examinations, cardiac evaluations, AαGlu activity assays in fibroblasts, urinary glucotetrasaccharide (HEX4) and blood creatine kinase (CK) analyses, and/or parental DNA analyses. Through these additional tests, patients with one or no known variants may be diagnosed with LOPD, potential LOPD, or non-LOPD (carrier or pseudodeficiency) [8].

## 3. AαGlu Enzyme Assay

In NBS for PD, AαGlu activity in DBSs is measured. Conventionally, the AαGlu activities in lymphocytes, fibroblasts, and skeletal muscles are analyzed for the diagnosis of PD [9]. Neutrophils in the blood contain maltase glucoamylase, another type of α-glucosidase. Because the pH at which this enzyme functions is consistent with that of AαGlu, the AαGlu activity assays in the blood are likely to result in false-negative results for defects in AαGlu [9]. However, large-scale NBS using DBSs has become possible owing to the use of acarbose, which inhibits the activity of maltase glucoamylase [10,11].

Measurement of AαGlu enzyme activity in DBSs can be carried out using fluorometry with the fluorogenic substrates of 4-methylumbelliferyl α-d-glucopyranoside (4MU-αGlc) [12], tandem mass spectrometry (MS/MS) [11], or digital microfluidic fluorometry [13]. Tandem mass spectrometry using mass-differentiated internal standards can quantify the corresponding enzymatic products and enables multiplex assays of a set of corresponding enzymes that cause lysosomal storage disorders (LSDs), such as PD, mucopolysaccharidosis (MPS), Fabry disease (FD), Gaucher disease (GD), Krabbe disease, and Niemann–Pick A/B disease. Additionally, digital microfluidics, a type of fluorometry, can be used to perform multiple assays of enzymes in the lysosome [14]. Tandem mass spectrometry is considered superior to fluorometry in terms of sensitivity, and some reports have shown that tandem mass spectrometry can distinguish pseudodeficiencies, which cannot be identified in patients with PD using fluorometry [15]. However, this is controversial because no reports have compared the outcomes of tandem mass spectrometry with those of fluorometry in the same samples. Moreover, the cutoff values used in NBS differ according to institution and region owing to factors such as differences in samples, measurement instruments, and humidity.

## 4. *GAA* Gene Analysis

*GAA* gene analysis is essential for definitive diagnosis of PD. The *GAA* gene is located on chromosome 17q25. It is 18.3 kb long, contains 20 exons, and encodes 952 amino acids. At present, more than 900 variants are registered in the ClinVar [16] or Pompe disease *GAA* variant databases [17], and the numbers are increasing. About two-thirds of these variants are classified by clinical significance and the other third of the variants are VOUS. Although bioinformatic tools such as Polyphen-2 [18], Human Splicing Finder [19], and Mutation Tester [20] are useful to estimate the pathogenicity of VOUS, these tools are insufficient for making the diagnosis. The progression of symptoms, treatment, and its outcomes are most important. Follow-up studies on the patients or potential patients with PD are essential and will help clinicians to diagnosis and determine proper treatment of patients with VOUS.

## 5. Pseudodeficiency

A pseudodeficiency allele is a change in the *GAA* gene sequence that results in AαGlu enzyme activity reduction, but is not enough to cause PD [21,22]. In our previous pilot program, the presence of pseudodeficiency alleles was shown to interfere with NBS for PD [23]. This suggests that NBS for PD must be able to distinguish PD cases and those with pseudodeficiency alleles in the *GAA* gene sequence. Asian patients are frequently homozygous or heterozygous for these pseudodeficiency alleles c.[1726G>A; 2965G>A] (p.[G576S; E689K]). Moreover, pseudodeficiency variants such as c. 1726G>A (p. G576S) are modifiers of pathogenic variants, which can result in greater reductions in GAA enzyme activity than with only the pathogenic variants [24]. Therefore, evaluation of pseudodeficiency in the *GAA* gene is essential in NBS.

Attempts have been made to distinguish cases with pseudodeficiency from patients with PD by methods other than *GAA* gene analysis; however, no reports have described a successful approach to achieve this goal.

## 6. AαGlu Activity in Fibroblasts

The measurement of AαGlu activity in fibroblasts was previously the gold standard for the diagnosis of PD [9]. This method can exclude the contribution of maltase glucoamylase activity. However, AαGlu assays in fibroblasts are difficult to perform as an NBS approach because acquiring fibroblasts for use in the AαGlu assay requires a skin biopsy, which is invasive and requires lengthy fibroblast culturing times. Despite this, AαGlu assays in fibroblasts are thought to be useful as additional tests for definitive diagnosis, even if *GAA* gene analysis cannot be used for the diagnosis of PD due to the presence of VOUS or only one known pathogenic variant for PD.

## 7. Cross-Reactive Immunological Material (CRIM)

Patients with IOPD should receive early ERT immediately after diagnosis of PD. Moreover, patients should undergo evaluation of CRIM before receiving ERT [25]. Patients with IOPD exhibiting residual AαGlu enzyme activity are CRIM-positive, and patients with IOPD exhibiting no residual AαGlu enzyme activity are CRIM-negative. CRIM-negative patients develop neutralizing antibodies for recombinant human lysosomal α-glucosidase (rhGAA) when receiving ERT, which impairs the effects of ERT [25,26].

In previous approaches for the evaluation of CRIM, residual AαGlu enzyme activity in fibroblasts was measured using an invasive method that required a long time to obtain results. Currently, the outcomes of *GAA* gene analysis contribute to estimations of the outcomes of CRIM [27].

## 8. ERT

ERT for patients with IOPD should be initiated as early as possible before irreversible damage occurs. Yang et al. indicated that early identification of patients with IOPD allows for very early initiation of ERT. Starting ERT even a few days earlier may lead to better patient outcomes [28]. Starting ERT early is effective in patients with LOPD as well; however, early ERT before presentation of signs or symptoms in patients with LOPD is generally not recommended [29]. Administration of ERT in the absence of symptoms of LOPD, even when blood CK and urine HEX4 are elevated, is avoided. The recommendation of ERT for patients with LOPD remains controversial.

Patients with LOPD should receive regular follow-ups, and levels of markers, such as blood CK and urine HEX4, should be monitored [30]. If patients with LOPD presenting symptoms of PD did not undergo NBS, their diagnosis and treatment are often delayed [31]. However, patients with LOPD diagnosed using NBS can be followed up and receive therapy immediately after presentation of the symptoms of PD [32]; treatment of these patients should not be delayed.

## 9. Immunomodulation to ERT

A variety of immunomodulation therapies have been developed to prevent the generation of neutralizing antibodies to rhGAA that would impair the effect of ERT. The immunomodulation therapies are elimination therapy for the neutralizing antibodies to rhGAA that have already been generated or prevention therapy for avoiding generation of the neutralizing antibodies before ERT. Rituximab, methotrexate, and intravenous immunoglobulins are often used for the immunomodulation therapies.

Prevention therapy has higher cost benefit than elimination therapy. Therefore, it is beneficial to identify CRIM-negative patients with IOPD before ERT in order to prevent generating neutralizing antibodies. Even some CRIM-positive IOPD patients are likely to develop neutralizing antibodies. The individualized T cell epitope measure scoring method, using a combination of individualized Human Leukocyte Antigen (HLA)-binding predictions and GAA genotype, may predict patient-specific risk of developing neutralizing antibodies to rhGAA [33].

Most patients with LOPD develop IgG antibodies to rhGAA, typically within 3 months of initiation of treatment [4]. Moreover, some patients who develop high, sustained antibody titers may have poorer clinical responses to treatment. Patients with IOPD or LOPD receiving ERT should routinely undergo tests for neutralizing antibodies for rhGAA. In rare cases in which high neutralizing antibodies interfere with the effects of ERT in patients with PD, we should consider the administration of immunosuppressants with ERT as well as discontinuing ERT.

## 10. The Follow-Up Period

Patients with IOPD who were diagnosed by NBS and received ERT should undergo regular follow-ups to assess treatment efficacy, onset of new symptoms, and deterioration of symptoms every month for the first 6 months or more [9,30]. In particular, cardiac evaluation is essential every month in the first 4 months of life and every 1–2 months thereafter. Clinicians should measure anti-rhGAA antibodies regardless of the state of CRIM. Patients with IOPD require immunosuppressants when they develop high titers of anti-rhGAA antibodies [34].

Patients with asymptomatic LOPD should undergo regular follow-up every 3 months during the first year after diagnosis. If they remain free of symptoms for 12 months, follow-ups every 3–12 months is required. Patients should receive ERT if they present signs or symptoms of PD.

Patients with symptomatic LOPD receiving ERT should undergo regular follow-ups monthly for 4 months after receiving ERT and then every 3 months thereafter, including monitoring for antibodies [30]. Because blood CK, aspartate aminotransferase (AST), alanine aminotransferase (ALT) levels, and urine HEX4 levels may increase before the onset of PD symptoms, these markers should be assessed regularly.

## 11. NBS Programs for PD Worldwide

The Newborn Screening Center at the National Taiwan University Hospital initiated an NBS program for PD in 2005. The outcomes of this large-scale NBS for PD in Taiwan demonstrated that the survival rates and ventilation-free rates of patients who were diagnosed with IOPD by NBS and received early ERT were higher than those of patients with IOPD who received ERT after presenting symptoms for IOPD [6,24]. Several regions in Japan also started NBS for PD in April, 2013 [7]. Moreover, the USA added PD to the Recommended Uniform Screening Panel (RUSP) and started NBS for PD in 2015. Currently, several countries worldwide have started pilot or regular NBS programs for PD (Table 1). The number and frequency of pseudodeficiencies and carriers between Japan and Taiwan are shown in Table 2 [35]; those in each country are displayed in Table 3.

### 11.1. Taiwan

Chien et al. performed the first large pilot NBS program to detect PD in newborns in Taiwan using a fluorometric enzymatic assay to determine AαGlu activity in DBSs. They conducted a pilot NBS of 132,538 newborns, accounting for almost 45% of newborns in Taiwan, between October 2005 and March 2007. Of the 132,538 newborns screened, 1093 (0.82%) underwent repeated DBS sampling, and 121 (0.091%) newborns were recalled for additional evaluation. PD was identified in 4 newborns (3 IOPD and 1 LOPD) [49]. Owing to this outcome, NBS for PD is now regularly conducted in Taiwan. Moreover, they identified 9 patients with IOPD and 19 patients with LOPD among 473,738 newborns by NBS for PD between October 2005 and December 2011 [36]. They launched a four-plex MS/MS LSD newborn screening test also including AαGlu (PD), acid α-galactosidase (FD), acid α-glucocerebrosidase (GD), and acid α-l-iduronidase (MPSI) in 2015. Through 2017, 64,148 newborns were screened for these four LSDs using their system. The cutoff levels in this new NBS system were established as 0.1 percentile of the population, or 13–15% of the normal mean. This NBS detected 20 infants with less than the cutoff value, and 1 patient with IOPD, 5 patients with LOPD, and 14 infants with false-positive results were identified [38].

Liao et al. reported the results of 191,786 newborns evaluated in an NBS program for PD using a system that could detect multiple LSDs by MS/MS from February 2010 to January 2013. After the initial DBS screening, 9 newborns were referred to hospitals directly with AαGlu values lower than the critical cutoff value (0.20 μmol/L/h) or combined with some clinical symptoms. In total, 874 (0.46%) newborns were recalled for second DBSs, 225 (0.12%) suspected newborns with decreased AαGlu activity were referred to hospitals, and 16 newborns were confirmed to have PD. In *GAA* gene analysis, 5 newborns were classified as IOPD and 11 newborns as LOPD. *GAA* gene analysis demonstrated that c.1935C>A (p.D645E) was detected in all cases of IOPD, c.[752C>T; 761C>T] (p.[S251L; S254L]) was detected in 8 cases of LOPD, and the c.1726G>A (p.G576S) pseudodeficiency variant was detected in 2 cases of IOPD and 5 cases of LOPD. The variants c.1840A>G (p.614A), c.2647-23delT, c.1054C>T (p.Q352*), IVS7+2T>C, and IVS17-5T>C were identified as novel variants. The false-positive rate in the tandem mass method was similar to that in fluorometric assays [37].

### 11.2. Japan

We started a pilot study of NBS for FD using 4-methylumbelliferyl-α-d-galactopyranoside (4MU-αGlc) in August 2006 and have conducted NBS of 5 LSDs, including PD, FD, GD, MPSI, and MPSII [50,51].

We reported the results of NBS for PD using 4MU-αGlc in 103,204 newborns [7]. Among these newborns, 225 newborns were retested using a second AαGlu assay, and 111 newborns with low AαGlu activities under the cutoff in the second AαGlu assay were evaluated for IOPD detection using physical and biochemical examinations (CK, ALT, AST, and lactate dehydrogenase), echocardiogram assessments, and a third AαGlu assay. For the 71 newborns with low AαGlu activity under the cutoff in the third AαGlu assay, *GAA* gene sequencing was performed using NGS. The AαGlu activities in fibroblasts were measured in 32 of the 71 newborns. In this study, no newborns developed IOPD, and 50 variants were detected. Eight variants were novel: c.547-67C>G, c.692+38C>T, c.1082C>A (p.P361Q), c.1244C>T (p.T415M), c.1552-52C>A, c.1638+43G>T, c.2003A>G (p.Y668C), and c.2055C>G (p.Y685*). The most common mutation was c.[752C>T;761C>T] (p.[S251L; S254L]), accounting for 20 alleles (14.1%, 20/142). The pseudodeficiency alleles c.1726G>A (p.G576S) and c.2065G>A (p.E689K) were detected in 71.8% (102/142) and 72.5% (103/142) of all newborns with low AαGlu activity, respectively.

This study identified 3 newborns with potential LOPD but without IOPD detection. Although these 3 patients developed no symptoms related to PD and received no treatment, the c.317G>A (p.R106H), c.1244C>T (p.T415M), and c.2003A>G (p.Y668C) mutations in these 3 patients were considered novel mutations. The prevalence of potential PD was 1 per 34,401 births. Newborns with both pseudodeficiency alleles and PD-associated pathogenic variants were detected in Japan as well as in Taiwan (Table 2). Appendix A displays the distribution of mutations and predictably pathogenic variants in NBS for PD in Japan.

### 11.3. USA

In 2008, the Advisory Committee on Heritable Disorders in Newborns and Children (ACHDNC) evaluated the NBS system for PD. The committee found significant evidence gaps related to the accuracy of screening and to the benefits and harms of presymptomatic diagnosis and precluded recommendation of NBS of PD for the Recommended Uniform Screening Panel (RUSP). In 2013, the ACHDNC reconsidered PD after it was nominated again. Based in part on new information presented to the ACHDNC by the external condition review workgroup, NBS for PD was recommended for addition to the RUSP and was added in March 2015. Prosser et al. estimated that screening 4 million babies born each year in the United States would detect 134 cases with PD including 40 cases with IOPD, compared with 36 cases detected clinically without screening [52]. NBS would also identify 94 cases of LOPD that might not become symptomatic for decades. By identifying 40 cases with IOPD, NBS would avert 13 deaths and identify 26 individuals requiring mechanical ventilation by the age of 36 months.

#### 11.3.1. Washington

In 2013, Scott et al. reported screening results for more than 110,000 newborns in Washington. They detected PD, FD, GD, MPSI, MPSII (α-l-iduronide-2-sulfatase), Niemann–Pick A/B disease (acid sphingomyelinase), MPSIV-A (galactose-6-sulfate sulfatase), MPSVI (N-acetylgalactosamine-4-sulfatase), and Krabbe disease (galactocerebrosidase) using a technology which simultaneously measured multiple enzyme activities by MS/MS [39]. AαGlu activity in DBSs was measured in 111,544 cases. A cutoff value was established as 15% of the mean value. Seventeen samples had enzyme activities with less than the cutoff value. Four cases (2 cases with the homozygous IVS1-13T>G variant, 1 case with the compound heterozygous c.365T>A/c.1925T>A variant, and 1 case with the compound heterozygous IVS1-13T>G/c.1-17C>T variant) were confirmed to be patients with LOPD or potential LOPD, 4 cases had a single nucleotide change on one allele (carrier of PD), 3 cases were identified as carriers with an additional pseudodeficiency allele, and 6 cases were heterozygotes for a pseudodeficiency allele only. The prevalence of infants with LOPD or potential LOPD was 1 per 27,800 births.

Elliott et al. evaluated 43,000 newborns in NBS using a multiplex MS/MS enzymatic activity assay of 6 lysosomal enzymes for PD, FD, GD MPSI, Niemann–Pick A/B disease, and Krabbe disease. The cutoff value was established as 10% of the mean value. A newborn with p.G576S/p.T602I (a probable low activity variant) and a newborn with the homozygous c.2168del13ins10 (p.A724Gfs*44) variant (a frameshift variant leading to a nearby stop codon) were identified [40].

#### 11.3.2. Missouri

A full-population pilot study of 43,701 newborns using DBSs in a multiplex fluorometric enzymatic assay for detecting PD, FD, GD, and MPSI was performed in Missouri on January 11, 2013 [41]. The cutoff values for AαGlu activities were set at the 0.17 percentile in the pilot study. Of the 18 cases that screened positive for AαGlu deficiency, 3 were diagnosed with IOPD, 3 were classified as LOPD, 2 were classified as potential LOPD or VOUS, 2 had pseudodeficiencies, and 3 were carriers. The prevalence of PD was 1 per 8740 births.

#### 11.3.3. Illinois

The Newborn Screening Laboratory of Illinois performed NBS for 5 LSD-associated enzymes, including PD, FD, MPSI, and Niemann–Pick A/B disease, among 219,973 newborn DBSs using MS/MS from November 1, 2014 to August 31, 2016 [42]. In total, 139 (0.06%) had a positive or borderline test result in PD, necessitating additional testing. The cutoff values for AαGlu activities were defined as follows: positive = less than or equal to 18% of the daily median value, and borderline = greater than 18% and less than or equal to 22% of the daily median value. Ten cases of PD (two cases of IOPD and eight cases of LOPD) were detected. The frequency of PD was 1 per 21,997 births. Two infants diagnosed with IOPD developed elevated CK levels and clear evidence of cardiomyopathy at the time of initial evaluation which included chest radiography, electrocardiography, and echocardiography. These patients are regularly receiving ERT. Eight infants diagnosed with LOPD had either homozygous or compound heterozygous variants of the common splicing mutation c.-32-13T>G observed in patients with LOPD. *GAA* pseudodeficiency was detected in 15 infants, including 14 identified as Asian (4 Chinese, 3 Filipino, 2 Korean, 2 Indian, 1 Japanese, and 2 others). There were four infants with an undetermined classification or “potential PD.”

#### 11.3.4. New York

A pilot NBS study for 18,105 newborns was performed through October 1, 2014 [43]. Six cases were positive in the screen (1 case of LOPD, 3 cases of pseudodeficiency, and 2 carriers) with a mean AαGlu activity of less than or equal to 15% of the daily mean activity, yielding a referral rate of 0.033%. One case with LOPD had the homozygous c.-32-13T>G variant and low leukocyte AαGlu activity; however, examination results and laboratory values were normal and HEX4 testing was negative. Two cases were homozygous for the common pseudodeficiency allele, and another case carried one copy of the pseudodeficiency allele; leukocyte AαGlu activity in these infants ranged from low to normal. One case was found to have c.2560C>T (p.R854*) and an intronic variant believed to be benign. This infant was diagnosed as being a carrier because AαGlu activity was high.

### 11.4. Austria

DBSs of 34,736 newborns were collected consecutively in the national routine Austrian NBS program from January 2010 to July 2010 and analyzed for enzyme activities of acid β-glucocerebrosidase, α-galactosidase, α-glucosidase, and acid sphingomyelinase by electrospray ionization MS/MS [44]. The cutoff value for AαGlu was based on the 0.1 percentile value from 5000 cases of AαGlu activity. This first-line screening for low AαGlu activity detected 25 cases, and retests showed 5 cases with low AαGlu activities. Sequence analyses of the *GAA* gene identified 4 cases with PD presenting homozygous c.896T>C (p.L299P), homozygous c.-32-13T>G, or compound heterozygous c.-32-13T>G/c.1551+1G>A (V480_I517del) variants. The prevalence of PD was 1 per 8684 births.

### 11.5. Hungary

In the Hungarian NBS program, 40,024 newborns were screened for PD, FD, GD, and Niemann–Pick A/B disease using MS/MS [45]. The 0.25th–0.5th percentile of AαGlu activities from 1000 cases were defined as the cutoff values; 663 cases (1.66%) were submitted for retesting. Among them, 163 cases (0.41%) had abnormal AαGlu activities in DBSs. After retesting, *GAA* gene analysis was performed in 64 (0.160%) cases with low AαGlu activity. *GAA* gene analysis detected 9 cases with PD and 25 carriers for PD. Three cases remained uncertain owing to inclusive *GAA* variants. Five cases had c.664G>A (p.V222M)/c.664G>A (p.V222M), and the other cases had c.-32-13T>G/c.-32-13T>G, c.664G>A (p.V222M)/c.2174G>A (p.R725Q), c.1216G>A (p.D406N)/c.1409A>C (p.N470T), and c.1552-3C>G/c.1552-3C>G. The 25 carriers had *GAA* gene variants related to PD, including -32-13T>G, c.307T>G (p.C103G), c.664G>A (p.V222M), c.763C>T (p.Q255X), c.841C>T (p.R281W), c.875A>G, c.1437+1G>A, c.1468 T>C (p.F490L), c.1552-3C>G, c.1903A>G (p.N635D), c.2237G>T (p.W746L), and c.2482-2A>G. The prevalence of PD was 1 per 20,012 births.

### 11.6. Italy

NBS programs for PD, FD, GD, and MPSI were performed using DBSs from 44,411 newborns by multiplexed MS/MS with the NeoLSD assay system in northeastern Italy from September 2015 to January 2017 [46]. Among the 44,411 newborns screened for the four LSDs, 40 cases (0.09%) had enzyme activity below 0.2 multiples of the median and were recalled for collection of a second DBS. Eight cases showed low AαGlu activities. Five patients underwent *GAA* gene analysis, and 2 cases were identified as juvenile types of PD because of the presence of the compound heterozygote of c.-32-13T>G (IVS1-13T>G)/c.236_246del (p.P79Rfs*12); elevated blood CK, AST, and ALT levels; and slightly enlarged heart findings. These patients underwent ERT immediately after diagnosis of PD. One case was classified as VOUS, and 2 cases were carriers for *GAA* variants. The frequency of PD was 1 per 22,205 births.

A total of 3403 newborns (1702 males and 1701 females) in the Umbria area of central Italy were screened for PD, FD, GD, and MPSI using DBSs [53]. The cutoff value was established as 35% of the normal median AαGlu activity. Although 3 cases showed AαGlu activities with less than the cutoff value in DBSs, none of these cases showed abnormal lymphocyte AαGlu activities. This pilot study identified no infants with PD.

### 11.7. Mexico

NBS for 6 LSDs, including PD, FD, GD MPSI, Niemann–Pick A/B disease, and Krabbe disease, using a multiplex MS/MS enzymatic assay was performed in 20,018 newborns (10,241 males and 9777 females) from July 1, 2012 to April 30, 2016 [47]. Nineteen presumptive positive infants showed low AαGlu activity in the first DBS. In the second AαGlu activity test, 16 infants were positive. Among these infants, 11 showed low leukocyte AαGlu activity. Five infants could not complete the protocol, three of whom lost healthcare insurance and 2 of whom refused to continue the protocol. In 11 infants with low leukocyte AαGlu activity, 10 infants harbored a pseudodeficiency variant associated with low AαGlu enzyme activity, but without signs of PD (homozygous or heterozygous for c.[1726G>A; 2065G>A] (p.[G576S; E689K])). Moreover, 2 infants harbored a compound heterozygous variant for the pseudodeficiency allele and the -32-13 T> G variant. The patient with potential LOPD presented with the c.1375G>A (p.D459N) variant, which was reported as an LOPD variant, and the VOUS c.1220A>G (p.Y407C) variant, which was predicted to be pathogenic.

### 11.8. Brazil

A pilot NBS study used a digital microfluidic platform to simultaneously measure the activities of AαGlu, acid β-glucosidase, α-galactosidase, and iduronidase to screen for PD, GD, FD, and MPSI in DBSs, respectively [54]. The cutoff value for AαGlu was defined as less than 30% of the mean enzyme activity in samples from 1000 unaffected babies. One case was identified to be pseudodeficiency, and no infants were identified with PD [48,55].

## 12. Genotype–Phenotype Correlation

Known variants in PD are registered in PD variant databases, including the Pompe Center (http://www.pompevariantdatabase.nl/pompe_mutations_list.php?orderby=aMut_ID1) and ClinVar (http://www.ncbi.nlm.nih.gov/clinvar). Moreover, in PolyPhen-2 (http://genetics.bwh.harvard.edu/pph2), the impact of missense variants on amino acid substitutions is evaluated. Human Splicing Finder (http://www.umd.be/HSF3/) can predict the impact of splicing abnormalities.

c.525delT (p.E176Rfs*45), a common frameshift variant in the Netherlands [56], c.1935C>A (p.D645E), a common missense variant in Taiwan [6], and c.2560C>T (p.R854*), a common nonsense variant in Africa [57], are considered variants for IOPD. c.-32-13T>G is commonly detected in Caucasian patients with LOPD, and patients with the homozygous c.-32-13T>G variant rarely develop cardiac manifestations as infants [58]. c.-32-13T>G is considered a variant in LOPD. However, because even patients with the same variants can develop both IOPD and LOPD phenotypes, the phenotype cannot be predicted from gene analysis alone. Moreover, the distribution of gene variants differs in each region. In the future, the accumulation of genetic information for patients in each region will be essential for predicting phenotypes.

## 13. Potential Concern of Screening for PD

NBS not only detects patients with IOPD but also patients with LOPD and potential LOPD. The effectiveness of NBS for identification of patients with LOPD and potential LOPD is controversial. Patients with LOPD who would present with PD symptoms within 2–3 years can receive medication before the progression of the symptoms, due to early diagnosis of LOPD through NBS. However, the demerits of NBS for patients with LOPD and potential LOPD should be considered because the time from diagnosis to presentation of PD symptoms may be more than 10 years, and some patients with LOPD or potential LOPD may not ever develop PD symptoms. Therefore, problems such as the psychological stress for the family due to LOPD and potential LOPD diagnosis [59], the cost and time of visiting hospitals and receiving medical examinations, and the potential of receiving overtreatment are issues likely to occur.

As shown in the guidelines for PD promulgated by the Pompe Disease Newborn Screening Working Group [30], clinicians should consider the need for psychosocial support for families during follow-ups for presymptomatic patients with LOPD. As more patients or potential patients with LOPD are diagnosed and followed up, it has been demonstrated that LOPD causes more symptoms than proximal myopathy or respiratory failure; it is a multiorgan disorder involving muscular, respiratory, musculoskeletal, peripheral nervous, vascular, cardiac, and gastrointestinal systems. Common symptoms reported in LOPD are proximal muscle weakness, trunk muscle weakness, exercise intolerance, shortness of breath, impaired cough, and gait difficulties. Because LOPD is a multisystemic disease, clinicians should be aware of all known symptoms and indicators in order to prevent delayed diagnoses and misdiagnoses. The understanding of the natural history of LOPD is advanced in the use of ERT. In the future, the disease concept of LOPD as well as IOPD will be more established.

## 14. Future Challenges

NBS programs for PD can contribute to early detection and early intervention in patients with IOPD and LOPD. Early ERT can change the natural clinical course and result in better outcomes in patients with IOPD. Because of changes in the natural clinical course, neurological manifestations, which had not previously been discussed, have become apparent [60]. For example, some patients with IOPD receiving ERT present with learning disorders as neurological manifestations. Currently available rhGAA therapy cannot cross the blood–brain barrier [61,62]. Moreover, patients with LOPD detected in NBS can receive follow-up and early intervention before exhibiting deterioration of PD symptoms [32].

Nevertheless, currently available NBS programs that evaluate AαGlu activity in DBSs, even those using tandem mass spectrometry, cannot discriminate pseudodeficiency from cases of PD or potential PD. Because families of newborns with pseudodeficiency have anxiety related to the results of NBS, new NBS programs, such as those using a combination of AαGlu enzyme assays and *GAA* gene analysis from DBS, can be used to distinguish pseudodeficiency from PD or VOUS. Such approaches are urgently needed.

In the future, next-generation treatments, including chaperon therapy [63] and gene therapy for PD [64,65], are expected to have favorable outcomes. Therefore, many researchers should contribute to the development of novel, improved NBS programs and the spread of such NBS programs to more regions around the world.

## Figures and Tables

**Figure 1 IJNS-06-00031-f001:**
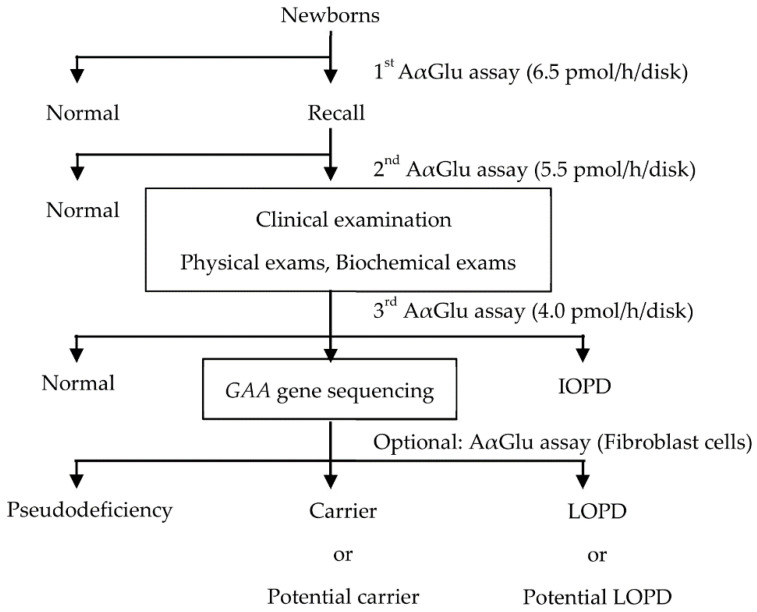
Flow chart of newborn screening (NBS) for Pompe disease (PD) in Japan. IOPD: infant-onset Pompe disease; LOPD: late-onset Pompe disease.

**Figure 2 IJNS-06-00031-f002:**
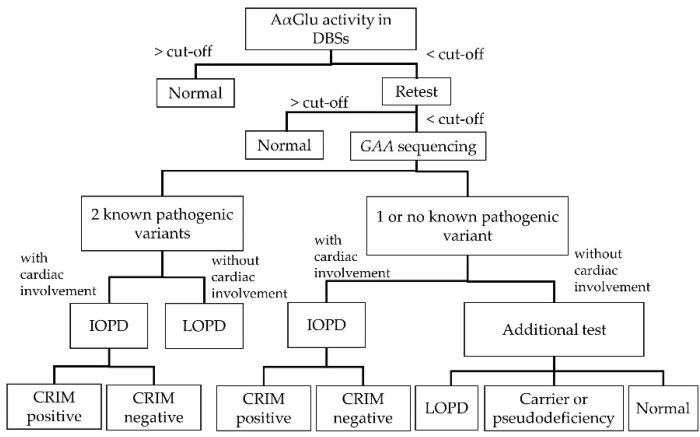
Flow chart of diagnosis for PD (modified from the Pompe Disease Newborn Screening Working Group [8]). DBS: dried blood spot; CRIM: cross-reactive immunological material.

**Table 1 IJNS-06-00031-t001:** Summary of NBS programs for Pompe disease.

Country (No. of Newborns)		No. of Recalls (%)	No. of Patients (Prevalence)		Screening Method	Frequently Detected Pathogenic Variants	Reference
			IOPD	LOPD			
Taiwan (473,738)		2210 (0.47)	9 (1/52,638)	19 (1/24,934)	fluorometric assay	c.1935C>A, c.2238G>C	Chiang et al. (2012) [36]
Taiwan (191,786)		874 (0.46)	5 (1/38,357)	11 (1/17,435)	MS/MS	c.1935C>A, c.[752C>T; 761C>T]	Liao et al. (2014) [37]
Taiwan (64,148)		92 (0.14)	1 (1/64,148)	5 (1/12,830)	MS/MS	NA	Chiang et al. (2018) [38]
Japan (103,204)		225 (0.24)	0	3 (1/34,401)	fluorometric assay	c.[752C>T; 761C>T], c.317G>A	Momosaki et al. (2019) [7]
USA	Washington (111,554)	17 (0.02)	0	4 (1/27,889)	MS/MS	c.-32-13T>G	Scott et al. (2013) [39]
	Washington (44,047)	9 (0.02)	0	1 (1/44,047)	MS/MS	c.2168del13ins10	Elliott et al. (2016) [40]
	Missouri (43,701)	18 (0.04)	3 (1/14,567)	5 (1/8740)	DMF	NA	Hopkins et al. (2015) [41]
	Illinois (219,973)	139 (0.06)	2 (1/149,987)	8 (1/37,497)	MS/MS	c.-32-13T>G	Burton et al. (2017) [42]
	New York (18,105)	6 (0.03)	0	1 (1/18,105)	MS/MS	c.-32-13T>G	Wasserstein et al. (2019) [43]
Austria (34,736)		5 (0.01)	0	4 (1/8684)	MS/MS	c.896T>C, c.-32-13T>G	Mechtler et al. (2012) [44]
Hungary (40,024)		163 (0.41)	0	9 (1/4447)	MS/MS	c.664G>A, c.-32-13T>G	Wittmann et al. (2012) [45]
Italy (44,411)		8 (0.02)	2 (1/22,206)	0	MS/MS	c.-32-13T>G, c.236_246del	Burlina et al. (2018) [46]
Mexico (20,018)		19 (0.09)	0	1 (1/ 20,018)	MS/MS	c.1375G>A	Navarrete-Martínez et al. (2017) [47]
Brazil (103,204)		NA	0	0	DMF	-	Bravo et al. (2017) [48]

NA: not available.

**Table 2 IJNS-06-00031-t002:** The distribution of pseudodeficiency alleles and PD-associated variants in newborns who were detected by NBS for PD.

Country/ Reference	Pseudodeficiency Alleles	No. of PD-Associated Variants			Prevalence (%)
		0	1	2	
Japan (*n* = 103,204)/ Momosaki, et al. (2019) [7]	Homozygous	24	8	0	32/71 (45.1%)
	Heterozygous	0	35	3	38/71 (53.5%)
	None	0	0	1	1/71 (1.5%)
	Diagnosis	Pseudodeficiency 24/71 (33.8%)	Carrier or potential carrier 43/71 (60.6%)	Patient or potential patient 4/71 (5.6%)	
Taiwan (*n* = 132,538)/ Labrousse et al. (2009) [35]	Homozygous	36	32	0	68/104 (65.4%)
	Heterozygous	0	27	7	34/104 (32.7%)
	None	0	0	2	2/104 (1.9%)
	Diagnosis	Pseudodeficiency 36/104 (34.6%)	Carrier or potential carrier 59/104 (56.7%)	Patient or potential patient 9/104 (8.7%)	

**Table 3 IJNS-06-00031-t003:** Number of pseudodeficiencies and carriers in each country.

Country (No. of Newborns)		Pseudodeficiency (with 0 PD-Associated Variants)		Carrier or Potential Carrier (with 1 PD-Associated Variant)		Reference
		No.	Genotype	No.	Genotype	
USA	Washington (111,554)	6	pseudodeficiency allele/wt. (*n* = 6)	7	pathogenic allele/wt. (*n* = 4) pathogenic allele/pseudodeficiency allele (*n* = 3)	Scott et al. (2013) [39]
	Washington (44,047)	0		0		Elliott et al. (2016) [40]
	Missouri (43,701)	2	NA	3	NA	Hopkins et al. (2015) [41]
	Illinois (219,973)	15	NA	19	NA	Burton et al. (2017) [42]
	New York (18,105)	3	c.[1726G>A; 2065G>A]/c.[1726G>A; 2065G>A] (*n* = 2) c.[1726G>A; 2065G>A]/wt.	2	c.2560C>T(VOUS)/c.858+20_858+26del*(Predicted Benign) c.[1726G>A; 2065G>A]/c.-32-13T>G	Wasserstein et al. (2019) [43]
Austria (34,736)		0		0		Mechtler et al. (2012) [44]
Hungary (40,024)		0		17	NA	Wittmann et al. (2012) [45]
Italy (44,411)		0		2	c.[1726G>A; 2065G>A]/c.-32-13T>G c.1726G>A/c.-32-13T>G	Burlina et al. (2018) [46]
Mexico (20,018)		8	c.[1726G>A; 2065G>A]/c.[1726G>A; 2065G>A] (*n* = 6) c.[1726G>A;2065G>A]/wt. (*n* = 2)	2	c.[1726G>A; 2065G>A]/c.-32-13T>G (*n* = 2)	Navarrete-Martínez et al. (2017) [47]
Brazil (103,204)		0		1	c.[1726G>A; 2065G>A]/c.-32-13T>G	Bravo et al. (2017) [48]

NA: not available.

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
