# Peer review of "Newborn Screening for Pompe Disease"

_2409-515X, 2020, doi:10.3390/ijns6020031_

Round 1

Reviewer 1 Report

This is a review article summarizing aspects of Pompe Disease relevant to the newborn screening public health and scientific community.  Such a review would be of interest and of use to this community.  Overall, I feel this article as submitted lacks sufficient depth in some key areas and inefficiently discusses the experiences of jurisdictions that have published or otherwise communicated their experience with screening for Pompe Disease.  Further, some key references are missing and some references appear to be misquoted.  As such, I feel the manuscript needs major revision before consideration for publication.

Issues

  • A major concern in many jurisdictions is the identification of many infants with LOPD when the primary target of screening is IOPD. Further, the challenges in determining whether an infant really has LOPD are not insubstantial.  There is potential harm in early identification of LOPD infants (e.g. psychosocial harms to parents, overtreatment with ERT which may not be required, the patient/system burdens of surveillance, etc.  This review does not adequately explain and summarize these issues.  Further, it would be helpful if the review could summarize the observed proportions of LOPD:IOPD in the screened populations, perhaps in the table (see below in my comments).
  • Better referencing of pivotal and subsequent treatment studies. The experience in IOPD has not been as positive as first envisioned. Likewise, treatment and treatment initiation is LOPD is more controversial than the paper suggests.
  • Immunomodulation – wrong references, Line 154/155 Cyclosporine and azathioprine not used, neutralizing antibodies being a major problem, both in CRM+ and -, should cite reference 31, tie into issues with ERT, need for prophylaxis vs rescue immunomodulation. Line 184 – “Patients with IOPD require immunosuppressants when they develop high titers of anti-rhGAA antibodies [31].” Immunomodulation perhaps should be its own section.  It is needed not only when high titres develop, but also prophylactically in CRM-ve patients.   This area of immune reactions in patients is a major issue in PD patient care and the topic is covered with brief sentences in different sections.  Reference 31 is a key citation to summarize information from.
  • Pseudodeficiency – spend more time on aggregating and reporting on the alleles noted in the programs/pilots listed. Provide a table.  Include a table of complex alleles.  If I were a reader from a screening lab looking for data to support implementation – this paper does not provide PD allele frequencies estimated from programs or published.
  • Line 38 – ERT is not essential for all patients. Some patients will opt for palliation.  Some late onset patients may not need treatment.  What are the risks of overtreatment?
  • Line 39-40 – this sentence applies to IOPD (the reference provided is limited to this population)
  • Line 43 – NBS is not optimal in that not all identified kids will be asymptomatic at the time of diagnosis, babies with late onset disease who may not develop symptoms until much later will be identified, treatment is not effective for all identified infants, etc.
  • Line 146 - Re fibroblast assays “However, our findings were not consistent with these previous assertions (that fibroblast assays may be useful to establish a diagnosis when molecular results are inconclusive) [7];”  The authors should briefly explain why.  Reviewing the paper cited, there is overlap between screen positive individuals with pseudodeficient genotypes and those are presumed carriers; as well as overlap with 2 possible LOPD patients who had fibroblast activity measured.  Assignment of LOPD was made on the basis of in silico interpretation of variant pathogenicity in these asymptomatic infants that were not followed for sufficient time to establish if they had PD.  There were no IOPD patients in the study.  I would suggest that this data is insufficient to call into question the diagnostic utility of fibroblast enzymology and either the concept should be removed from this review or the issues should be explicitly explained.
  • Line 171 – the reference provided is regarding autophagy and does not provide supporting evidence for the statement that “administration of ERT when blood CK and urine HEX4 levels increase is effective in patients with no symptoms of LOPD”. Nor is this recommended in the important reference 28 (which also does not reference evidence to support the above assertion).  I note that Prof Nakamura is an author on the latter paper and this may be a mis-referencing in this submission.
  • Survey of pilots and current program experiences. It would be preferable if up to date data could be presented.  g. NY state data is from their 2014 small pilot, but they would have accumulated much more experience since then.  While additional work for the authors, and while not all jurisdictions would be willing to share newer information, this would be very helpful to include in a review published in IJNS.
    • Table: Include columns for recall rate, screen positive rate (to allow understanding of false-positive rate), and carriers. Consider adding columns for pseudodeficiency variants observed and complex alleles observed, or adding a second table if data on frequency of such alleles can also be added.  The most common LOPD variants observed in each region would also be helpful.  Finally, I would also recommend changing the columns for IOPD and LOPD prevalence to IOPD and LOPD # of cases (prevalence).  So the Taiwan row would be 9 (1/52,638) and 19 (1/24,934) for these two columns.  This will make it easier for the reader to quickly appreciate the relative birth prevalence observed for the two types of PD.
    • I would also suggest removing descriptions of the particular variants, allele frequencies, etc from the text and into a table.

Taiwan experience (Lines 220-234)

The numbers don’t add up.  “Four cases with IOPD and three cases with LOPD were identified. In total, 874 (0.46%) newborns were recalled for second DBSs, 225 (0.12%) suspected newborns with decreased AαGlu activity were referred to hospitals, and 16 newborns were confirmed to have PD. In GAA gene analysis, five newborns were characterized as IOPD, and 11 newborns were classified as LOPD.”

Further, the issue of proportion of late onset to infantile onset cases being identified by NBS programs is problematic aspect of PD screenings and needs to be better highlighted in this review.

GAA gene analysis demonstrated that c.1935C>A (p.D645E) was detected in all cases of IOPD (70% of the 10 mutated chromosomes), c.[752C>T; 761C>T] (p.[S251L; S254L]) was detected in eight cases of LOPD (45% of the 22 mutated chromosomes), and the c.1726G>A (p.G576S) pseudodeficiency variant was detected in two cases of IOPD and five cases of LOPD (25% of the 32 mutated chromosomes). The variants c.1840A>G (p.614A), c.2647-23delT, c.1054C>T (p.Q352*), IVS7+2T>C, and IVS17-5T>C were identified novel variants. The false-positive rate in the tandem mass method was similar to that in fluorometric assays.  The use of the term “mutated chromosomes” is not accepted terminology and I find it difficult to understand this section (this comment generally applies to the survey of pilots and current programs).  I think the authors are referring to the alleles found in the described patients.  Some of which seem to be complex alleles with more than one variant [pathogenic or pseudodeficient] on the same allele).  But the numbers don’t easily add up – e.g. 25% of 32 mutated chromosomes is referring to the 32 alleles found in the 16 patients.  7 patients (2 cases of IOPD and 5 cases of LOPD) must at least one complex allele, and 1 patient had 2?  Is this correct? This should be made more clear in the paper.  Further, if the proportion of complex alleles is this high, this is quite notable and important to highlight to the NBS community who may be considering reporting policies for cases identified with pseudodeficiency variants and deserves more attention in the paper.

  • Finally, there are some additional references that could be added. In particular a key reference that should be cited, and information included in the review is:

Newborn Screening for Pompe Disease: Synthesis of the Evidence and Development of Screening Recommendations Alex R. Kemper, Wuh-Liang Hwu, Michele Lloyd-Puryear and Priya S. Kishnani Pediatrics November 2007, 120 (5) e1327-e1334

Author Response

Reviewer 1:

This is a review article summarizing aspects of Pompe Disease relevant to the newborn screening public health and scientific community. Such a review would be of interest and of use to this community.  Overall, I feel this article as submitted lacks sufficient depth in some key areas and inefficiently discusses the experiences of jurisdictions that have published or otherwise communicated their experience with screening for Pompe Disease.  Further, some key references are missing and some references appear to be misquoted.  As such, I feel the manuscript needs major revision before consideration for publication.

(Author’s response)

We are grateful for your consideration and review. We have responded to each comment, point-by-point.

Issues

・A major concern in many jurisdictions is the identification of many infants with LOPD when the primary target of screening is IOPD. Further, the challenges in determining whether an infant really has LOPD are not insubstantial. There is potential harm in early identification of LOPD infants (e.g. psychosocial harms to parents, overtreatment with ERT which may not be required, the patient/system burdens of surveillance, etc. This review does not adequately explain and summarize these issues. Further, it would be helpful if the review could summarize the observed proportions of LOPD:IOPD in the screened populations, perhaps in the table (see below in my comments).

(Author’s response)

Thank you for your advice. As you mention, determining whether an infant really has LOPD is difficult. Particularly, in Japan, patients with IOPD are less detected than in Taiwan. However, patients with potential LOPD is detected as same as in the other countries. The benefit and harms about detecting patients with LOPD is controversial. Therefore, we added the issue of detecting LOPD in NBS in the section of 13. Potential Concern of Screening for PD. Moreover, we added the table showing the the observed proportions of LOPD:IOPD in the screened populations (Table 1)

・Better referencing of pivotal and subsequent treatment studies. The experience in IOPD has not been as positive as first envisioned. Likewise, treatment and treatment initiation in LOPD is more controversial than the paper suggests.

(Author’s response)

Thank you for your advice. We modified the section of ERT as following.

“Early ERT starting is effective in patients with LOPD also; however, early ERT before presentation of signs or symptoms in patients with LOPD is generally not recommended [29]. Administration of ERT in the absence of symptoms of LOPD even when blood CK and urine HEX4 are elevated is avoided. The indication of ERT for patients with LOPD remains controversial.”

  • Immunomodulation – wrong references, Line 154/155 Cyclosporine and azathioprine not used, neutralizing antibodies being a major problem, both in CRM+ and -, should cite reference 31, tie into issues with ERT, need for prophylaxis vs rescue immunomodulation. Line 184 – “Patients with IOPD require immunosuppressants when they develop high titers of anti-rhGAA antibodies [31].” Immunomodulation perhaps should be its own section. It is needed not only when high titres develop, but also prophylactically in CRM-ve patients. This area of immune reactions in patients is a major issue in PD patient care and the topic is covered with brief sentences in different sections.  Reference 31 is a key citation to summarize information from.
  • (Author’s response)

Thank you for your advice. Following your advice, we made the section of CRIM and Immunomodulation (9. Immunomodulation to ERT) and described them in this section. We removed the wrong reference.

  • Pseudodeficiency – spend more time on aggregating and reporting on the alleles noted in the programs/pilots listed. Provide a table.  Include a table of complex alleles.  If I were a reader from a screening lab looking for data to support implementation – this paper does not provide PD allele frequencies estimated from programs or published.(Author’s response) 
  •  
  • Thank you for your advice. Following your advice, we added the table showing the frequency of pseudodeficiency and complex allele (Table 2 and 3).
  •  
  • Line 38 – ERT is not essential for all patients. Some patients will opt for palliation.  Some late onset patients may not need treatment.  What are the risks of overtreatment?
    Line 39-40 – this sentence applies to IOPD (the reference provided is limited to this population)
  • (Author’s response) 
  • Thank you for your advice. We deleted the content of ERT for LOPD.
  • Line 43 – NBS is not optimal in that not all identified kids will be asymptomatic at the time of diagnosis, babies with late onset disease who may not develop symptoms until much later will be identified, treatment is not effective for all identified infants, etc.
  • (Author’s response) 
  • Thank you for your advice. Following your advice, because NBS is benefical for detecting IOPD, we applied this content for IOPD only.
  • Line 146 - Re fibroblast assays “However, our findings were not consistent with these previous assertions (that fibroblast assays may be useful to establish a diagnosis when molecular results are inconclusive) [7];” The authors should briefly explain why.  Reviewing the paper cited, there is overlap between screen positive individuals with pseudodeficient genotypes and those are presumed carriers; as well as overlap with 2 possible LOPD patients who had fibroblast activity measured.  Assignment of LOPD was made on the basis of in silico interpretation of variant pathogenicity in these asymptomatic infants that were not followed for sufficient time to establish if they had PD.  There were no IOPD patients in the study.  I would suggest that this data is insufficient to call into question the diagnostic utility of fibroblast enzymology and either the concept should be removed from this review or the issues should be explicitly explained.
  • (Author’s response) 
  • Thank you for your advice. As you mention, we can not detect IOPD in our screening. We removed this description in this review article.
  • Line 171 – the reference provided is regarding autophagy and does not provide supporting evidence for the statement that “administration of ERT when blood CK and urine HEX4 levels increase is effective in patients with no symptoms of LOPD”. Nor is this recommended in the important reference 28 (which also does not reference evidence to support the above assertion).  I note that Prof Nakamura is an author on the latter paper and this may be a mis-referencing in this submission.
  • (Author’s response)
  • Thank you for your advice. Following your advice, we modified the description as following.

“Early ERT starting is effective in patients with LOPD also; however, early ERT before presentation of signs or symptoms in patients with LOPD is generally not recommended [29]. Administration of ERT in the absence of symptoms of LOPD even when blood CK and urine HEX4 are elevated is avoided. The indication of ERT for patients with LOPD remains controversial.”

  • Survey of pilots and current program experiences. It would be preferable if up to date data could be presented.  g. NY state data is from their 2014 small pilot, but they would have accumulated much more experience since then.  While additional work for the authors, and while not all jurisdictions would be willing to share newer information, this would be very helpful to include in a review published in IJNS.(Author’s response)  Thank you for your advice. We added the revised table following your advice (Table 1).
    • I would also suggest removing descriptions of the particular variants, allele frequencies, etc from the text and into a table.Thank you for your advice. Unfortunately, we could not include the particular variants, allele frequencies, etc into a table, and left the description. Taiwan experience (Lines 220-234) Thank you for your advice. We are sorry. We removed the sentence that “Four cases with IOPD and three cases with LOPD were identified.Further, the issue of proportion of late onset to infantile onset cases being identified by NBS programs is problematic aspect of PD screenings and needs to be better highlighted in this review. GAA gene analysis demonstrated that c.1935C>A (p.D645E) was detected in all cases of IOPD (70% of the 10 mutated chromosomes), c.[752C>T; 761C>T] (p.[S251L; S254L]) was detected in eight cases of LOPD (45% of the 22 mutated chromosomes), and the c.1726G>A (p.G576S) pseudodeficiency variant was detected in two cases of IOPD and five cases of LOPD (25% of the 32 mutated chromosomes). The variants c.1840A>G (p.614A), c.2647-23delT, c.1054C>T (p.Q352*), IVS7+2T>C, and IVS17-5T>C were identified novel variants. The false-positive rate in the tandem mass method was similar to that in fluorometric assays.   Thank you for your advice. We removed the description of “mutated chromosomes”.I think the authors are referring to the alleles found in the described patients. Some of which seem to be complex alleles with more than one variant [pathogenic or pseudodeficient] on the same allele).  But the numbers don’t easily add up – e.g. 25% of 32 mutated chromosomes is referring to the 32 alleles found in the 16 patients. 7 patients (2 cases of IOPD and 5 cases of LOPD) must at least one complex allele, and 1 patient had 2? Is this correct? This should be made more clear in the paper. Further, if the proportion of complex alleles is this high, this is quite notable and important to highlight to the NBS community who may be considering reporting policies for cases identified with pseudodeficiency variants and deserves more attention in the paper.(Author’s response) 
    • Thank you for your advice. Following your advice, we added the table showing the frequency of pseudodeficiency and complex allele (Table 2 and 3).
    • (Author’s response)
    • The use of the term “mutated chromosomes” is not accepted terminology and I find it difficult to understand this section (this comment generally applies to the survey of pilots and current programs). 
    •  
    • (Author’s response)
    • The numbers don’t add up.  “Four cases with IOPD and three cases with LOPD were identified. In total, 874 (0.46%) newborns were recalled for second DBSs, 225 (0.12%) suspected newborns with decreased AαGlu activity were referred to hospitals, and 16 newborns were confirmed to have PD. In GAA gene analysis, five newborns were characterized as IOPD, and 11 newborns were classified as LOPD.”
    •  
    • (Author’s response)
  •  
  • (Author’s response)
  • ・Table: Include columns for recall rate, screen positive rate (to allow understanding of false-positive rate), and carriers. Consider adding columns for pseudodeficiency variants observed and complex alleles observed, or adding a second table if data on frequency of such alleles can also be added.  The most common LOPD variants observed in each region would also be helpful.  Finally, I would also recommend changing the columns for IOPD and LOPD prevalence to IOPD and LOPD # of cases (prevalence).  So the Taiwan row would be 9 (1/52,638) and 19 (1/24,934) for these two columns.  This will make it easier for the reader to quickly appreciate the relative birth prevalence observed for the two types of PD.
  • Thank you for your advice. We added this paper in the reference in our review article.
  •  
  • Finally, there are some additional references that could be added. In particular a key reference that should be cited, and information included in the review is: Thank you for your advice. The author in the above recommended paper by you changed the notion for Newborn Screening for Pompe Disease in the following paper. 
  • Prosser, L.A.; Lam, K.K.; Grosse, S.D.; Casale, M.; Kemper, A.R.; Workgroup, on behalf of the C.R.; Children, A.C. for H.D. in N. and Using Decision Analysis to Support Newborn Screening Policy Decisions: A Case Study for Pompe Disease. MDM policy Pract. 2018, 3.
  • We added this paper in the reference [42] of our review article and described the content in the USA section.
  • (Author’s response)
  • Newborn Screening for Pompe Disease: Synthesis of the Evidence and Development of Screening Recommendations Alex R. Kemper, Wuh-Liang Hwu, Michele Lloyd-Puryear and Priya S. Kishnani Pediatrics November 2007, 120 (5) e1327-e1334

Reviewer 2 Report

Several areas mentioned in the text are very important to countries/states and all who are to embark on NBS for Pompe Disease.  Your pointing to the importance of frequent follow-up of infants diagnosed with LOPD in order to not miss onset of symptoms very important.  Also, the importance of neurologic issues in infants treated with enzyme replacement therapy, which doesn't cross the BBB is important moving forward and will likely press for more infants receiving gene therapy when properly studied and available.

Author Response

Reviewer 2: Several areas mentioned in the text are very important to countries/states and all who are to embark on NBS for Pompe Disease.  Your pointing to the importance of frequent follow-up of infants diagnosed with LOPD in order to not miss onset of symptoms very important.  Also, the importance of neurologic issues in infants treated with enzyme replacement therapy, which doesn't cross the BBB is important moving forward and will likely press for more infants receiving gene therapy when properly studied and available.

(Author’s comment)

Thank you for your review. We are grateful for your comment.